# How Firms Can Improve Sustainable Performance on Belt and Road Initiative

**Tao Zhao [1,†], Jung-Mo Koo [2,†] and Min-Jae Lee [3,*]**

1 School of Management, Suzhou University, No. 1769 Xuefu Avenue, Suzhou 234000, China
2 Department of Business Administration, Mokwon University, Daejeon 35349, Korea
3 Department of International Trade & Logistics, Mokwon University, Daejeon 35349, Korea
* Correspondence: mjlee@mokwon.ac.kr
† These authors contributed equally to this work.

**Abstract:** This study investigates the digitalization capabilities and the moderating effect of green open innovation (GOI) that firms need to achieve triple bottom line (TBL) performance in the Belt and Road Initiative (BRI). This study explores the structure of business ecosystems that firms need to achieve sustainable performance and investigates open innovations that can be promoted based on them. The data used in the analysis was collected from 474 manufacturing firms pursuing partnerships among ecosystem participants to promote Sustainable Development Goals (SDGs) in the BRI. The moderating regression analysis is used in this study. We found that digitization capabilities (DCs) have a significant effect on a firm's TBL performance. In addition, it was confirmed that GOI has a positive moderating effect on digitalization capabilities and a firm's economic performance. Based on these results, we also believe our model contributes to the current knowledge by filling several research gaps, and our findings offer valuable and practical implications not only for achieving sustainable growth but also for the creation of competitive advantage.

**Keywords:** digitalization capabilities; green open innovation; triple bottom line performance; business ecosystem; Belt and Road Initiative

## 1. Introduction

There is a growing perception that sustainability plays an important role in a firm's competitive advantage [1,2]. As the COVID-19 pandemic and climate change come as unexpected threats to the survival of firms, the issue of sustainability is becoming a keyword in management decisions. Firms are focusing their resources on seeking capabilities and strategies to increase sustainability for survival [3,4]. Sustainability is a concept that meets current demand without infringing on future generations' opportunities to meet demand [5]. A firm's sustainability is based on implementing these concepts and focuses on three principles: social sustainability, environmental sustainability, and economic sustainability [6–8]. These three dimensions, or pillars, represent the 'triple bottom line (TBL)' to examine a firm's performance and impact [9,10]. Some scholars, including [11], argue that firms can create and grow value by focusing on these three factors to secure sustainability.

Meanwhile, among the many interesting changes facing firms in pursuing sustainability, the most powerful and important issue is how to understand and create the new technology revolution [12,13]. In other words, due to the development of digital technologies, the boundaries between industries are becoming increasingly obscure, and the business environment has been rapidly altered. Therefore, as firms that improve process efficiency and systematically manage resources based on digital technology appear [14,15], digitalization capabilities are attracting the attention of scholars and practitioners as a driving force for achieving competitive advantage [16]. Such a situation forces firms to discard old mindsets, think outside the box, and accept even disruptive changes. In this vein, research on digitization capabilities (DCs) has made significant progress in strategic

and technical management research areas [17–19], but it does not provide a structured perspective on how and why firms use DCs to enhance sustainability. Moreover, while existing research emphasizes the DCs of firms to achieve competitive advantages by finding hints from a resource-based view (RBV) [20–22], some argue that domestic and multinational firms that invest in business ecosystems, providing a venue for both cooperation and competition, may need to understand better the open innovation required by other sources of knowledge. In particular, some point out that in order to increase sustainability, firms should strengthen open innovation that pursues green innovation [23–25]. For instance, Roh, et al. [26] found that a green open innovation (GOI) approach helps the sustainability of the business ecosystem based on an empirical analysis of Korean manufacturing firms. As such, enabling GOI through DCs is an important way for manufacturing firms to gain a competitive advantage in the digital economy. However, existing research on the development of digital transformation and innovation capabilities in the manufacturing industry is based on conceptual and review studies, which makes it insufficient to explore the mechanisms of DCs that enable GOI. Moreover, while the need for digital transformation has emerged as a stepping stone to enhancing a firm's sustainability, it has become urgent to secure dynamic capabilities for it, but it has not satisfactorily explored its capabilities and strategies to successfully achieve sustainable performance of a firm. In this sense, our study investigates how DCs can achieve sustainable performance for firms participating in the Belt and Road Initiative (BRI), the most widely recognized business ecosystem, and it examines the moderating effect of GOI.

The BRI is a business innovation ecosystem created to improve the globalization, regional cooperation, and economic growth of countries located along the overland Silk Road Economic Belt and the Maritime Silk Road. Urgent initiatives to encourage sustainable consumption and production are required in the area of present global climate governance and objectives for carbon neutrality [27,28]. On this account, firms participating in the BRI are inevitably induced to enhance sustainable performance, including cleaner production, and form a coopetitive climate regardless of the host country [29,30]. In this respect, firms in the BRI may face institutional pressures to promote coopetition among ecosystem participants with the aim of sustainable development and growth, and firms are expected to strengthen their interactions with external partners to ensure business legitimacy.

In this atmosphere of the BRI, we argue that GOI strategies will play an important role in creating sustainable performance. In an open and cooperative environment, it is more effective for a firm to combine strengths with other participants to create value (or performance) than to work alone as a strategic actor to achieve competitive advantage. In order to achieve sustainable performance in the BRI, firms will consider GOI strategies for green innovation. GOI can strengthen the sustainability of the business ecosystem and immediately improve the competitive edge of a firm by promoting cooperation and sharing among ecosystem participants. Chesbrough and Appleyard [31] argued that open innovation can improve the sustainability of the business ecosystem by allowing firms to embrace ideas from external partners and provide research and development (R&D) results to other participants. Lichtenthaler [32] pointed out that open innovation can help firms reduce R&D costs, increase customer acceptance, and accelerate innovation to achieve a competitive advantage. Meanwhile, countries along the BRI are in the middle or low stages of the global value chain and are under enormous pressure from environmental pollution and carbon emissions [29,33]. Therefore, firms in the BRI enhance sustainability by strengthening cooperation with ecosystem participants to promote green innovation that can change their business models environmentally [26]. As such, discussions on the creation of a sustainable firm through GOI in the BRI are actively underway. Based on the above prior studies, we expect firms to actively develop DCs, strengthen GOI with business ecosystem participants, and increase sustainability on issues that cannot be solved by a single firm alone (e.g., climate change, poverty, and inequality) in the BRI.

The expected contributions of this study are as follows. First, we propose an integrated framework that includes DCs and GOI that can solve social and environmental issues as well as achieve economic value to enhance a firm's sustainability, focusing on the BRI. Second, our study demonstrates the importance of GOI strategies for a firm's sustainable growth in the face of growing social demands for clean development.

## 2. Theoretical Background and Hypotheses Development

### 2.1. BRI as a Coopetitive Climate Business Ecosystem

The theoretical basis of our analysis is the business ecosystem perspective. A business ecosystem is where the actors who make up the ecosystem sometimes compete but cooperate and share resources to create added value and evolve jointly [34]. In a business ecosystem, interdependence, coexistence, coevolution, competition, cooperation, value-sharing, and value-creation among ecosystem participants have emerged. Iansiti and Levien [35] defined a business ecosystem as a group in which a number of agents remained loosely connected and interdependent for survival and the reinforcement of a competitive advantage. Cohesive firms creating a business ecosystem can cooperate with various business participants; members may achieve a competitive advantage based on interactions with other participants, allowing a standalone firm to pursue challenging achievements within a boundary. Furthermore, since some achievements are associated with social and environmental problems, firms attempt to solve these issues more effectively by activating a business ecosystem with participants [36]. For that reason, close accordance and cooperation among participants in a business ecosystem are required.

The BRI focuses on social and environmental issues and strives to foster cooperation and competition (i.e., coopetition) among participants and enhance co-evolution to create an innovative business ecosystem that meets sustainable development goals (SDGs) [30]. At the same time, the Chinese government is strengthening its institutions and support to build a robust, eco-friendly, and inclusive innovation system for the BRI [37]. In the midst of these situations, the BRI presents opportunities to encourage renewable energy and green industry growth (e.g., the Green Silk Road Fund, and the Belt and Road Green Investment Fund) and promotes coopetition among participants [38]. As a facilitator of the SDGs, the BRI offers solutions to address global economic imbalances, low-carbon development, and shaky alliances [39]. Accordingly, we contend that the BRI is an innovative business ecosystem with the most significant potential to advance global economic integration, creating a coopetitive climate among participants to address infrastructure gaps, social inequality, and climate change.

Under the circumstances of a coopetitive climate business ecosystem, we can expect the following impacts: First, competition within the business ecosystem can be converted from a zero-sum game to a positive-sum game [40,41]. A coopetition atmosphere in the ecosystem can move toward a win-win status since it allows participants (i.e., competitors and cooperators) to open and share their capabilities and resources [42]. As needed, participants open and share resources with competitors and innovate their business model by complementing mutual shortcomings and relocating their knowledge, thereby increasing the size of the entire pie in the market [43,44]. Second, the business ecosystem enables resource utilization efficiency to achieve sustainable performance [45,46]. With social actors essential to solving social problems, collaborative reciprocity encourages participants' interactions and improves resource efficiency to reach common goals.

In the BRI as a coopetitive climate, for sustainable growth of firms, various competencies are needed to promote interaction among ecosystem participants, but our study selected DCs as important for this.

### 2.2. Digitalization Capabilities and 'Triple Bottom Line' Performance

This study defines DCs as the capability to improve or build the functions and processes necessary to promote digital transformation by utilizing digital technology and resources [16,47]. DCs can theoretically stimulate the original resources that are potentially

present in the enterprise, free up new resources owned by external firms, and facilitate their innovation by coordinating and integrating all internal and external resources [48]. In addition, DCs digitally transform business processes such as customer experience and business operations, which essentially changes the business ecosystem by affecting members and their networks [49]. Based on their DCs, firms are showing signs of increasing productivity, optimizing resource use, and achieving sustainable performance in the business ecosystem [50,51]. At the same time, they are digitally transforming their business models. For instance, manufacturing firms can leverage digital technologies (e.g., big data, IoT, CPS, and blockchain) to extensively analyze and evaluate their internal production processes to increase the sustainability of their manufacturing activities [52,53]. DCs based on digital technologies enable manufacturing firms to efficiently manage machine usage and energy requirements and increase sustainability [54]. In other words, DCs help firms make their processes more efficient and manage their resources better, which helps them grow in a way that is sustainable from an economic, social, and environmental point of view.

Meanwhile, scholars and practitioners have not reached an agreement on the concept and measurement of a firm's sustainable performance [55,56], but there is considerable agreement that sustainability consists of three dimensions: environmental, social, and economic sustainability [6–8]. Elkington [9] argued that based on this TBL, it was possible to investigate the sustainable performance and impact of the firm. Environmental sustainability can be defined as upholding or improving the integrity of the Earth's life-supporting systems [57]. It is to secure long-term protection of the business ecosystem and create environmental values to minimize the artificial impact of the natural world. Social sustainability refers to the social value of sustainability related to relationships with various stakeholders as well as the impact on the social system operated by the organization [58,59]. It addresses issues such as regional imbalances, polarization, and a fair distribution of opportunities across and within generations. Norton and Toman [60] defined economic sustainability as an organization's impact on the economic conditions of its stakeholders and on economic systems at local, national, and global levels. In order to achieve sustainable performance in corporate management, business activities should be carried out in a way that protects the environment, is socially viable, and is economically sound [24]. We argue that leveraging DCs to conserve energy and natural resources is a highly productive way to achieve value in terms of TBL.

In this vein, several studies, including Dubey, et al. [61], show a linear relationship between digital technologies and capabilities and these three aspects of sustainable performance. According to Shivajee, et al. [62], the use of digital tools improves the manufacturing process and enables clean production. Beier, et al. [63] explained that digitalization has socio-technical technologies in which economic, social, and organizational opportunities converge. Chaudhuri, Subramanian and Dora [20] confirmed that firms have a positive impact on the circular economy by efficiently utilizing large amounts of information and creating environmental value through digital capabilities. Piyathanavong, et al. [64] also pointed out that awareness and investment in digital technologies are needed to improve green and cleaner performance. Moreover, some scholars stress that digital capabilities can provide clues to solve ethical and sustainable supply chain problems [65,66]. As such, research examining the sustainability of firms through digital technology and capabilities focused on the environmental, social, and economic values arising from digitalization and examined their impact.

In particular, the Chinese government is accelerating the digital transformation of the BRI with the prolonged COVID-19 Pandemic, the absence of global leadership, the rise of the digital economy, and the demand of developing countries. In a similar vein, at the 2017 BRI International Cooperation Summit, President Xi Jinping proposed the Digital Silk Road and pledged to collaborate with BRI participants, particularly developing countries, in ICT infrastructure construction and digital economic development. Thus,

**Hypothesis 1.** *The DCs positively affect the TBL performance of a firm in the BRI.*

### 2.3. Moderating Effect of Green Open Innovation

Open innovation plays a critical role in promoting sustainability and efficiency in the advancement of green innovation [26]. Undoubtedly, GOI contributes to sustainability, but doing and implementing it requires large-scale restructuring beyond existing technologies and processes, which is a significant challenge for firms. Open innovation offers a number of benefits to firms. Open innovation processes include multiple internal and external technology sources and multiple internal and external technology commercialization channels [67,68], which allow firms to open up the innovation process in two directions [69]. Inbound open innovation (IOI) allows firms to benefit from a combination of new ideas and knowledge, new market opportunities, and new problem-solving capabilities [70]. Outbound open innovation (OOI) enables firms to leverage existing knowledge and skills to gain financial and non-monetary benefits while minimizing obsolescence threats and maintaining competitiveness [70]. Most previous studies on open innovation's effects suggest that open innovation has a positive effect on various measures of a firm's performance. Some studies, including Reed, et al. [71] empirically proved that open innovation practices have a positive and significant effect on firms' profitability. Chesbrough and Di Minin [72] described the positive effects of open innovation in achieving positive social change. Li-Ying, et al. [73] empirically demonstrate that environmental innovation is positively influenced by open innovation. To pursue sustainable values, firms redefine interactions, priorities, and behaviors with stakeholders and make extensive adjustments to improve business processes. But firms can't be sustainable if they treat economic benefits, environmental conservation, and social responsibility as separate things. Instead, they need to work together with ecosystem participants through collaboration, cooperation, and co-creation. This can't be done without an open innovation approach [70,74].

Open innovation research suggests that firms' strategic behavior to open up innovation strategies is influenced by both internal and external factors [70]. The literature agrees on the fact that capabilities development and evolutionary processes are dependent on the business context [62,69]. Consistent with this view, previous studies focusing on the external context characteristics of open innovation suggest that opening up innovation strategies is more suitable in business environments characterized by globalization, competitive intensity, and market and technological turbulence [75–77]. Chesbrough and Crowther [70] explains that firms should actively seek opportunities from outside by scanning the external environment and comparing the knowledge needed for strategic goals before starting innovative projects, including R&D. Laursen and Salter [78] found that the scope of use of external knowledge is partially shaped by environmental factors such as environmental turbulence or technological opportunities. In order to maintain competitiveness and create sustainable value in this rapidly digitalized, dynamic business environment, firms adopt open innovation practices to expand mutual exchange with outside participants and acquire knowledge and technologies [79]. In this sense, the way firms pursue innovation capabilities to effectively develop GOI is expected to depend on the business ecosystem climate and the performance they want to achieve.

Meanwhile, the BRI emphasizes open multilateralism based on coopetition among participants to promote green development and sustainability of the business ecosystem [80]. In addition, cooperation between firms is being strengthened in areas such as the digital economy and AI with the goal of building the digital Silk Road, and digital technology exchanges are promoted in connection with the green development of the BRI [81]. Firms in the BRI are strengthening their DCs as a core competency to achieve sustainable performance (i.e., TBL performance) in this business ecosystem climate, and are being seen pursuing GOI to effectively develop them [82–84]. For example, IT firms such as Alibaba, Tencent, and Huawei are contributing significantly to sustainable issues (e.g., reducing carbon emissions, income polarization, and regional imbalances, etc.) by opening their AI platforms and programs for free and establishing an environment where offline firms can easily access inventory management and smart logistics services. Moreover, Xiaomi shares digital platforms for free and incorporates startups into their value chains. Based

on this, it realizes social responsibility through fostering startups. Using the preceding considerations as justification, we suggest the following hypothesis:

**Hypothesis 2.** *GOI (inbound and outbound) interacting with DCs works as a moderator for TBL performance of a firm in the BRI.*

### 3. Methodology

#### 3.1. Sample and Data Collection

Figure 1 is our suggested research model. We conducted an empirical analysis with a dataset of manufacturing firms in the BRI to test our hypotheses. The BRI is a business ecosystem that is designed to strengthen partnerships among participants and realize sustainable growth in order to promote economic development along the Silk Road route. All this provided a new platform for China's continued investment development and had a significant impact on resolving regional imbalances and driving the qualitative growth of the economy. In particular, the BRI is focusing on creating a business ecosystem where participants lead green growth, promote digitalization, and coexist based on joint development with underdeveloped regions. President Xi Jinping emphasizes the importance of the BRI in responding to the global crisis through strengthening multilateralism and building an open and sustainable business ecosystem. In this vein, the Chinese government highlights coopetition among economic agents to encourage firms to make more social contributions and solve environmental problems as well as economic affluence. This concept is consistent with BRI philosophy of pursuing common prosperity based on cooperation and competition with ecosystem participants.

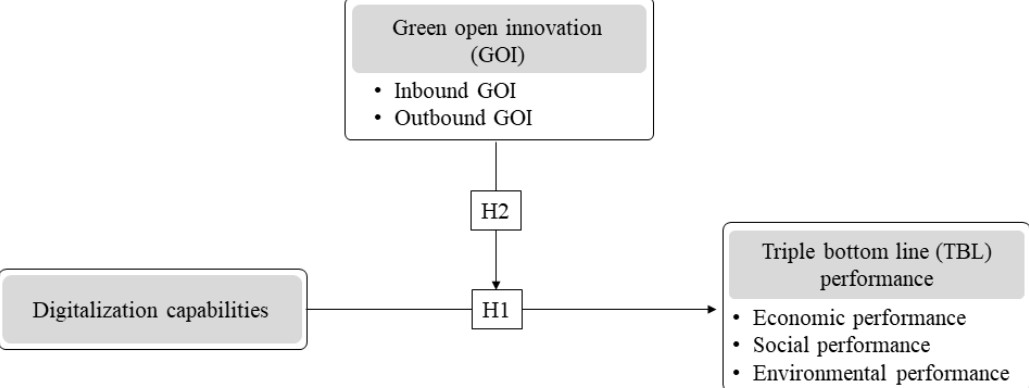

**Figure 1.** Research model.

We selected multiple BRI regions in China. According to China's Chamber of Commerce and Industry, more than 100,000 firms are pushing for collaboration projects with economic agents (i.e., firms, government institutes, and universities) to achieve SDGs. We collected data from manufacturing firms working on joint projects with economic players to achieve SDGs in the BRI. We selected a random sample of 1000 firms from the sampling lists obtained regarding firms' industry-university collaboration projects provided by the Provincial Chamber of Commerce and Industry. We executed the survey method with the support of a new research organization in the local Chinese market to entice survey participants to answer and increase the response rate. As a result of our data collection processes, we received 474 usable questionnaires, implying a 47.4 percent effective response rate.

#### 3.2. Variables and Measurement

In this study, all the dependent, independent, and moderating variables were assessed using multiple-item, five-point Likert scales, where 1 = strongly disagree and 5 = strongly agree. The dependent variable in this study was the firm's TBL performance. We inquired about a firm's economic (or financial), environmental, and social performance to mea-

sure this performance through each of the three questions [85,86]. Specifically, economic performance asks about a firm's net profit, sales growth, and market share, which have increased compared to competitors over the past five years. Social performance has prompted firms to respond to their level of contribution to alleviating inequality, strengthening social safety, and solving overall social problems over the past five years. Environmental performance has asked about the level at which firms over the past five years have contributed to reducing carbon emissions, reducing resources (energy), and solving overall environmental issues.

The independent variable of this study is digitalization capabilities. To measure digitalization capabilities, respondents' perceptions were measured through four items on how firms used digital technology [87,88]. Specifically, we asked whether a firm aims for digitalization; whether a firm realizes information exchange through digitalization; whether a firm build a network using digital technology; and whether a firm collects data using digital sources.

The moderating variable in this study is GOI. In this study, GOI was measured in two aspects: inbound green open innovation (IGOI) and outbound green open innovation (OGOI), referring to Chesbrough [89] and Roh, Lee and Yang [26]. Each question was asked with four items. IGOI asked about licensing out related to green, selling green technologies and knowledge to the outside, having dedicated organizations to commercialize green-related intellectual property (sales, cross-patents, and decentralization), and exchanging green technologies and knowledge to the outside. OGOI has allowed firms to respond to external ideas about technologies (and patents) from the outside, external ideas to create green value, systems to acquire and explore green-related intellectual property, and exchanges with external partners.

This study used firm size and age as control variables [90,91]. This study controlled firm size because large firms have access to more or better functions than small firms, while small firms can have more flexibility and the ability to innovate faster. We measured the number of a firm's employees in five ranges. In addition, firm age was included as a control variable because it could affect sustainable growth and performance.

## 4. Analyses and Results

### 4.1. Validity Test

Prior to verifying the hypothesis, this study conducted a validity analysis to find out how accurately the survey data measures the concepts required for hypothesis verification. After the above process, the hypothesis verification was verified through moderated regression analysis. Table 1 presents the results for validity. The validity analysis reviewed the 'convergent validity' and 'discriminant validity' of variables through common factor analysis and then confirmed their reliability through the Cronbach's alpha coefficient. As a result of factor analysis, digitization capabilities were classified into a single dimension; GOI, which is a control variable, was classified into IGOI and OGOI; and TBL performance, which is a dependent variable, was classified into economic, social, and environmental performance. Based on this, the Cronbach's alpha coefficient of all constituent concepts analyzed was 0.70 or higher (0.830 < all alpha coefficients < 0.915). In addition, the variance inflation factors (VIF) values of all measured variables were less than 5 (1.317 < all VIF values < 3.831), which confirmed there was no problem with multicollinearity. Moreover, the correlations and descriptive statistics for the variables are presented in Table 2. Therefore, it can be determined that the validity of the variables used in this study has been secured.

**Table 1.** Results of validity analysis.

| Variable | Indicators | Component 1 | 2 | 3 | 4 | 5 | 6 | Cronbach's $\alpha$ |
|---|---|---|---|---|---|---|---|---|
| Digitalization capabilities (DCs) | DC1 | 0.901 | | | | | | |
| | DC2 | 0.829 | | | | | | 0.915 |
| | DC3 | 0.925 | | | | | | |
| | DC4 | 0.916 | | | | | | |
| Inbound green open innovation (IGOI) | IGOI1 | | 0.789 | | | | | |
| | IGOI2 | | 0.869 | | | | | 0.864 |
| | IGOI3 | | 0.861 | | | | | |
| | IGOI4 | | 0.851 | | | | | |
| Outbound green open innovation (OGOI) | OGOI1 | | | 0.731 | | | | |
| | OGOI2 | | | 0.811 | | | | 0.830 |
| | OGOI3 | | | 0.858 | | | | |
| | OGOI4 | | | 0.845 | | | | |
| Economic performance (EP) | EP1 | | | | 0.906 | | | |
| | EP2 | | | | 0.882 | | | 0.877 |
| | EP3 | | | | 0.899 | | | |
| Social performance (SP) | SP1 | | | | | 0.918 | | |
| | SP2 | | | | | 0.915 | | 0.911 |
| | SP3 | | | | | 0.932 | | |
| Environmental performance (EN) | EN1 | | | | | | 0.902 | |
| | EN2 | | | | | | 0.897 | 0.860 |
| | EN3 | | | | | | 0.851 | |

**Table 2.** Descriptive statistics and correlations.

| Variables | Mean | SD | 1 | 2 | 3 | 4 | 5 | 6 | 7 | 8 |
|---|---|---|---|---|---|---|---|---|---|---|
| 1. DCs | 4.167 | 0.723 | 1 | | | | | | | |
| 2. IGOI | 4.086 | 0.679 | 0.660 *** | 1 | | | | | | |
| 3. OGOI | 3.931 | 0.777 | 0.596 *** | 0.703 *** | 1 | | | | | |
| 4. EP | 4.241 | 0.673 | 0.686 *** | 0.704 *** | 0.594 *** | 1 | | | | |
| 5. SP | 4.057 | 0.715 | 0.659 *** | 0.674 *** | 0.578 *** | 0.651 *** | 1 | | | |
| 6. EN | 3.953 | 0.758 | 0.561 *** | 0.661 *** | 0.540 *** | 0.630 *** | 0.570 *** | 1 | | |
| 7. Firm size | 3.023 | 1.651 | 0.195 *** | 0.151 ** | 0.137 ** | 0.183 *** | 0.040 | 0.120 ** | 1 | |
| 8. Firm age | 2.430 | 1.129 | 0.123** | 0.121 ** | 0.060 | 0.166 *** | 0.053 | 0.097 * | 0.650 *** | 1 |

Note: n = 474, * $p < 0.05$, ** $p < 0.01$, *** $p < 0.001$.

### 4.2. Bias Testing

In this study, the dependent and independent variables were subjectively measured by the same person at the same time. In this case, the answer itself might contain the respondent's bias, which implies the possibility or risk of common method bias. Thus, we verified whether standard method bias was applied or not by performing a one-factor analysis before conducting a full-scale statistical analysis. According to Podsakoff, et al. [92], "One of the most widely used techniques that have been used by researchers to address the issue of common method bias is what has come to be called Harman's one-factor (or single-factor) test" (p. 889). We entered all variables measured subjectively by the respondents into this testing method. The results showed that six factors were divided, and the largest factor was 43.34%, which suggests that common method bias was not a concern in this study. According to Podsakoff, MacKenzie, Lee and Podsakoff [92], the presence of a substantial number of common methods should be suspected in cases where (1) a single factor emerges from the factor analysis or (2) the largest factor accounts for the majority of the covariance among the measures (i.e., more than 50%).

*4.3. Hypothesis Testing*

Table 3 presents the results of the regression analysis effect for hypotheses verification. First, Model 1, Model 4, and Model 7 showed the effect of DCs, which is an independent variable, on the dependent variable. The independent variable of this study was found to have a positive relationship with all dependent variables. Model 1 was explained by 47.7% (adjusted $R^2$ = 0.474), Model 4 was explained by 43.3% (adjusted $R^2$ = 0.440), and Model 7 was explained by 31.5% (adjusted $R^2$ = 0.311). Therefore, hypothesis 1 was supported. Second, Model 2, Model 5, and Model 8 explain the effect of independent variables, including GOI, on dependent variables, and each explanatory power was 58.7% (adjusted $R^2$ = 0.578), 55.2% (adjusted $R^2$ = 0.547), and 46.8% (adjusted $R^2$ = 0.463). Third, Model 3, Model 6, and Model 9 showed a moderating effect, including interaction variables, and were found to have some significant moderating effects. Specifically, it was found that IGOI and OGOI between DCs and economic performance had a significant positive moderating effect. In social and environmental performance, however, there was no significant moderating effect. Model 3 was explained at 60.9% (adjusted $R^2$ = 0.603), Model 6 was explained at 55.3% (adjusted $R^2$ = 0.548), and Model 9 was explained at 47.3% (adjusted $R^2$ = 0.465). Therefore, Hypothesis 2 was partially accepted.

**Table 3.** Results of moderated regression analysis.

| Variables | Model 1 | Model 2 | Model 3 | Model 4 | Model 5 | Model 6 | Model 7 | Model 8 | Model 9 |
|---|---|---|---|---|---|---|---|---|---|
| Dependent Variable | Economic Performance | | | Social Performance | | | Environmental Performance | | |
| DCs | 0.676 *** (0.629) | 0.362 *** (0.337) | 0.562 *** (0.523) | 0.677 *** (0.669) | 0.370 *** (0.366) | 0.453** (0.448) | 0.559 *** (0.585) | 0.198 *** (0.208) | 0.107 (0.112) |
| IGOI | | 0.384 *** (0.380) | 1.405 *** (1.393) | | 0.359 *** (0.378) | 0.305 (0.321) | | 0.460 *** (0.513) | 0.805 ** (0.898) |
| OGOI | | 0.105 * (0.091) | −0.854 ** (−0.740) | | 0.121 ** (0.111) | 0.293 (0.269) | | 0.099 * (0.097) | −0.429 (−0.419) |
| DCs*IGOI | | | 1.806 *** (−0.240) | | | 0.093 (0.013) | | | −0.605 (−0.091) |
| DCs*OGOI | | | 1.530 *** (0.195) | | | −0.660 (−0.037) | | | 0.845 (0.121) |
| Firm size | −0.003 (−0.001) | −0.007 (−0.003) | 0.005 (0.002) | −0.125 ** (−0.054) | −0.129 ** (−0.056) | −0.129 ** (−0.056) | −0.012 (0.005) | −0.015 (−0.007) | −0.013 (−0.006) |
| Firm age | 0.085 (0.051) | 0.073 (0.044) | 0.071 (0.042) | 0.051 (0.032) | 0.041 (0.026) | 10.009 (0.026) | 0.036 (0.024) | 0.020 (0.014) | 0.019 (0.013) |
| $R^2$ | 0.477 | 0.587 | 0.609 | 0.443 | 0.552 | 0.553 | 0.315 | 0.468 | 0.473 |
| $R^2$ adjusted | 0.474 | 0.578 | 0.603 | 0.440 | 0.547 | 0.548 | 0.311 | 0.463 | 0.465 |
| F | 142.875 *** | 135.704 *** | 103.494 *** | 124.707 *** | 115.415 *** | 82.347 *** | 72.095 *** | 82.637 *** | 59.794 *** |

Note: Unstandardized coefficients with standard errors in parentheses. * $p < 0.05$, ** $p < 0.01$, *** $p < 0.001$.

## 5. Discussion and Conclusions

This study expands the sustainable management literature by empirically evaluating the moderating effect of GOI on the DCs and TBL performance of firms in the BRI from a business ecosystem perspective. This is an informative finding because research on DCs in the BRI has been limited, and no research has been conducted on the moderating role of GOI to date. In terms of application, the results of this study verified the moderating effect of GOI, provided useful advice to managers to help firms in the BRI achieve TBL performance, and presented meaningful implications for policymakers to achieve the goal of the BRI.

*5.1. Theoretical Implications*

The theoretical framework of this study is valuable in expanding understanding of GOI activities by empirically verifying the DCs of a firm's sustainability in the BRI as a coopetitive climate business ecosystem. From the point of view of the business ecosystem theory and open innovation, our study offers a new channel to explain how DCs affect TBL performance. Our empirical results targeting manufacturing firms in the BRI show

evidence of the argument that DCs play a positive role in improving economic, social, and environmental performance. Empirical evidence typically supports the theoretical arguments of some studies that firms will create economic benefits, solve social issues, and protect the natural environment based on the utilization of digital technology and digital transformation based on them. In this light, we find that manufacturing firms doing business in the BRI have significant opportunities for strategic solutions to meet the sustainable development pursued by the BRI in DCs. With the recent spotlight on improving firm sustainability [93,94], our finding of a significant positive relationship between DCs and TBL performance improvements in the BRI is very encouraging.

Another considerable contribution of our study is the discovery of the moderating role of GOI. Our results show that GOI significantly strengthens the relationship between DCs and economic performance. This result is consistent with the open innovation perspective. Some researchers, such as Kennedy, et al. [95], argued that open innovation based on R&D and collaboration projects could generate sustainable value [96,97]. However, in this study, no significant effects were found in social and environmental performance other than economic performance. For the green growth of the business ecosystem, public spending on education, research, and development is important for firms to smoothly promote open innovation. In this vein, Zhang, et al. [98] pointed out this lack of activity in the BRI. In addition, many firms in emerging markets use it to improve economic performance rather than implement open innovation to develop environmental and social values [99]. In other words, firms in emerging markets focus on economic performance as they need to form strategic alliances with international partners and engage in costly innovation activities to overcome the shortcomings of emerging markets, such as lack of resources and capabilities [100,101]. Our findings indicate that firms driving GOI can achieve better economic performance based on their DCs. It is a meaningful discovery that reinforces the theoretical argument that it is important to pursue open innovation in strengthening DCs and enhancing economic performance in the BRI.

### 5.2. Managerial Implications

This study conveys meaningful implications for both practitioners and policymakers. First, our findings show that DCs have a positive effect on a firm's TBL performance improvement in the BRI. These results reveal the importance of DCs to improving sustainability through the utilization of digital technologies and digital transformation to solve economic, social, and environmental issues. In particular, with the rapid spread of digitalization in the business environment due to the fourth industrial revolution, the importance of utilizing DCs in the pursuit of sustainability for firms is increasing, and we propose the justification to secure them. Second, our results show that GOI strengthens the relationship between DCs and economic performance. Thus, firms should focus on securing DCs, strengthen open innovation to effectively acquire these dynamic capabilities, and actively participate in enhancing sustainability through coopetition among ecosystem participants. Meanwhile, the BRI aims to build a sustainable ecosystem through the Green Silk Road and the Digital Silk Road, and the results of our research conducted on firms in the BRI provide useful and practical advice to BRI policymakers to activate and supplement it. They should develop support policies (e.g., supporting expensive green innovation activities and R&D) that are more effective in creating ecosystems to create environmental and social values, which can bring sustainability and competitive advantage to firms in the BRI.

### 5.3. Limitations and Future Research Directions

Our study has three limitations as follows: First, this study focuses on exploring the moderating role of GOI. Some studies [102,103] have shown that closed innovation is important to ensure sustainability in dynamic and uncertain environments, such as digitization. Future research should identify open and closed innovation together and investigate their impact on corporate sustainable performance. Second, our sample consists

of manufacturing firms doing business activities in China, which may raise concerns about the possibility of generalization of GOI in the BRI. In the future, the generalizability and empirical results of the framework can be checked by expanding the sample to include firms from other BRI countries and regions. Third, it is also worth noting that a firm's organizational culture has a significant influence on promoting open innovation [104–106]. Confirming the GOI and a firm's performance according to the organizational culture can be a good way to increase the sustainability of a firm in the BRI.

**Author Contributions:** T.Z. and M.-J.L. contributed to the conceptualization, methodology, investigation and writing—original draft. J.-M.K. and M.-J.L. performed research model, data collection, data curation and formal analysis. T.Z., J.-M.K. and M.-J.L. participated in the manuscript revision, review, editing and validation. All authors have read and agreed to the published version of the manuscript.

**Funding:** This research received no external funding.

**Institutional Review Board Statement:** Not applicable.

**Informed Consent Statement:** Not applicable.

**Data Availability Statement:** Not applicable.

**Acknowledgments:** The authors would like to thank the editors and anonymous reviewers for their insightful comments and suggestions.

**Conflicts of Interest:** The authors declare no conflict of interest.

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
