# Peer review of "How Firms Can Improve Sustainable Performance on Belt and Road Initiative"

_sustainability, doi:10.3390/su142114090_

Round 1

Reviewer 1 Report

The theme of the article is very current, important and pertinent to science, being consistent with the journal's objective. The article complies with the journal's instructions. However, I would just like to raise a few points:

1. The title seems long and redundant with the key words and hypotheses formulated;

2. In the abstract I would not put acronyms;

3. In view of the existing acronyms, I would also put an acronym for the expression "digitalization capabilities";

4. Line 91: R&D acronym, without the respective meaning (appears on line 245);

5. Line 195: coherence in author referencing; iden line 367;

6. Line 210 - H1, I would only place: The digitalization capabilities positively affect theTBL performance of a firm in the BRI.

7. Line 269 - H2, I would only place: Green open innovation (inbound and outbound) interacting with digitalization capabilities works as a moderator for TBL performance of a firm in the BRI.

8. In the body of the text there is no reference to figure 1. However, I would suggest placing it in point 3. Methodology;

9. I suggest checking the references (e.g. DOI).

Author Response

We would like to thank the reviewer for the insightful and constructive comments, which have helped to refine the paper in many ways. We list below the actions we have taken as a result of the review. Please see the attachment. 

Reviewer 2 Report

The authors appear to have conducted an ambitious study. Did this study not receive any funding? The authors must describe the complete process of data collection in detail because the process described by the authors is too complex and may take a lot of time and money. When did this survey begin, and when did it end? Which research institution or commercial survey firm assisted in its completion? Also, please provide the complete survey questionnaire as an appendix, not just the abbreviated questions in the main text.

Author Response

(The authors gave the same response as above.)

Reviewer 3 Report

The article explores an interesting, current, pertinent and relevant object of study, not only for the academy but for the management of organizations, for which I congratulate the authors for the work carried out.

Literature review is balanced, presenting the conceptual model (supported in the literature), the methodological and operational process of the investigation is presented, as well as the statistical process that allowed the testing of hypotheses.

However, the authors do not satisfactorily explore issues related to digital transformation (industry 4.0), sustainability and sustainable development goals - SDG, as well as the analysis of dynamic capabilities (author: David J. Teece) to face the digital transition and sustainability.

In the same way, I believe that the authors should also present the problem and the research question that guided the investigation.

Although the research instrument and the measurement variables are described in point 3.2, I suggest that the authors present the questionnaire used (or a table that systematizes the dimensions and items) at the end of the paper.

Good luck

Author Response

(The authors gave the same response as above.)
